# Gender and Pharmacists’ Career Satisfaction in the United States

**DOI:** 10.3390/pharmacy9040173

**Published:** 2021-10-21

**Authors:** Manuel J. Carvajal, Ioana Popovici, Patrick C. Hardigan

**Affiliations:** 1Department of Sociobehavioral and Administrative Pharmacy, College of Pharmacy, Nova Southeastern University, 3200 South University Drive, Fort Lauderdale, FL 33328-2018, USA; ip153@nova.edu; 2Public Health Program, Dr. Kiran C. Patel College of Allopathic Medicine, Nova Southeastern University, 3200 South University Drive, Fort Lauderdale, FL 33328-2018, USA; patrick@nova.edu

**Keywords:** career satisfaction, gender disparities, job-related preferences, job satisfaction, pharmacist workforce

## Abstract

Job satisfaction reflects pharmacists’ evaluation of their current work experiences, while career satisfaction is an evaluation of how satisfied pharmacists are with their profession across various jobs. The objectives of this article were to measure career satisfaction and specific facets of current-job satisfaction of U.S. pharmacists, compare satisfaction across genders, and examine the determinants of career satisfaction. This study was based on self-reported survey data collected from a random sample of licensed pharmacists practicing throughout the United States. The sample consisted of 422 men and 315 women. Within each gender, pharmacists’ career satisfaction was modeled using ordinary least squares as a function of three sets of variables: personal characteristics, earnings and workweek, and other job-related variables. Female pharmacists exhibited higher levels of contentment with their careers than their male counterparts. Their career-satisfaction levels were not affected by age, marital status, annual earnings, or average workweek, covariates that systematically influenced male pharmacists’ career satisfaction. Job satisfaction substantially affected pharmacists’ long-term career satisfaction. Male and female pharmacists responded differently to stimuli, so a uniform set of work-related incentives may not be effective for both genders. Initiatives perceived by male practitioners as increasing satisfaction may be adversely perceived by female practitioners, and vice-versa.

## 1. Introduction

Professional satisfaction may be conceptualized as the level of contentment with one’s chosen occupation and the activities performed as part of that occupation’s responsibilities. Fulfillment occurs when a person’s work expectations are aligned with his/her environment [1] and his/her abilities, values, interests, and needs interact synergistically [2], yielding high quality-of-worklife levels [3,4]. Such interaction may be experienced in the short run regarding a specific position or place of employment, in which case the degree of contentment is considered to be job satisfaction, or in the long run, possibly over the course of more than one job, in which case it is labeled as career satisfaction.

Job satisfaction reflects pharmacists’ comprehensive evaluation of their current work, encompassing the mixture of their experiences in a position and/or setting. It is a global abstraction that allows analysts to measure a pharmacist’s or group of pharmacists’ aggregate wellbeing generated from a job at a given time. Practitioners are asked to assess their work-related conditions, often by means of comparison groups; their responses are influenced by intrinsic (i.e., job is interesting, challenging, meaningful, etc.) and extrinsic (i.e., pay and benefits, job security, advancement opportunities, etc.) forces, and it is up to the pharmacist to decide the relative importance of each facet of his/her employment [5,6].

Career satisfaction is a concept similar to job satisfaction, with two variations: (1) it embraces a longer-term perspective and (2) the spectrum of facets being considered in the assessment transcends a specific position/setting and refers to broader professional activities. It is a retrospective evaluation of how satisfied practitioners are with their choice of, and experience within, pharmacy as a profession across the various jobs they have held [7,8]. Thus, career satisfaction entails a fusion of both past and present considerations, whereas job satisfaction is restricted to the present. While research on job satisfaction has produced substantial empirical evidence, research on career satisfaction has been scant and has lacked a comprehensive view [9,10,11,12,13]. Exploring the influence of current-job experiences on the longer-term assessment of career satisfaction was of special interest here.

Within a human-capital framework, this article sought to:Measure the levels of career satisfaction and specific facets of current-job satisfaction of U.S. pharmacists,Compare the levels of satisfaction between male and female practitioners,Design and test a model depicting the levels of career satisfaction of each gender as a function of selected covariates, andExplore within each gender the effect of selected facets of current-job satisfaction on career satisfaction.

These pursuits are important because the effectiveness of any organization largely depends on the motivation of its employees, and motivation is determined by professional satisfaction [14,15]. Pharmacists’ satisfaction with their work and professional activities affects not only their employers but also their own wellbeing and their quality of patient care [16,17]. Several studies have found that higher levels of professional satisfaction are associated with less absenteeism and fewer intentions to change jobs or leave the profession, as well as fewer medication errors [18].

## 2. Methods

This study was based on self-reported survey data. Participants were asked to assess their career satisfaction (one question) using a 0–10 intensity scale, with 10 denoting the greatest level of satisfaction. This scale provided more room for discrimination in participants’ response than is normally provided by a Likert scale, yet the range of options ensured the adequacy of the mean as a measure of central tendency; in other words, there was no possibility of outliers. This type of indicator has been applied successfully in previous studies [19,20]. Also included in the questionnaire were requests for information on workweek hours and wage-and salary earnings, which had not been probed by other pharmacist workforce studies.

Male and female pharmacists were analyzed separately because several studies suggest that they respond differently to socioeconomic stimuli in the configuration of their workforce behavior [21,22], with women showing greater levels of satisfaction with their current jobs than men [23,24]. Within each gender, pharmacists’ career satisfaction was formulated as a function of three sets of variables: personal characteristics, earnings and workweek, and other job-related variables. Separate functions consisting of identical covariates were designed and tested using ordinary least squares to compare the direction, magnitude, and statistical significance of the coefficients on career satisfaction in the presence of all other covariates, thus avoiding disparities that might be attributed to interaction effects. An alternative model using a dummy variable for gender was discarded because of its likely incorrect assumption that the responses to covariates were equal for men and women [25,26]; in fact, gender differences in these responses, if they existed, were at the crux of the empirical evidence pursued in this article.

### 2.1. Data

The survey data were gathered from returns to a questionnaire sent to 2400 registered pharmacists practicing throughout the United States. These pharmacists were selected by Medical Marketing Services (MMS) using a simple random sampling scheme. MMS is a leading provider of lists of U.S. pharmacists and other healthcare professionals; its data depository is drawn from practitioners representing all fields within the profession. Approximately 90% of the estimated 281,560 pharmacists practicing at the time [27] were included in the MMS data files.

The survey questionnaire, previously validated and exclusively prepared for this and other workforce studies [28], was mailed by the authors in March 2012; a reminder was sent two weeks later. The sample size was chosen according to Cochran’s formula developed for categorical and other outcomes [29], with a 5% sampling error. This research effort was supported solely by internal university funds, and Institutional Review Board approval was secured to conduct the probe.

### 2.2. Covariates

Two personal-characteristics variables were identified: age and marital status. Older pharmacists tend to be more satisfied with their work than younger pharmacists [17,23,30,31]. They seem to experience a reduction in their aspirations, along with their satisfaction gap, as they grow older and realize that they have limited options in the workforce [32,33]. Although pharmacists may experience high levels of satisfaction shortly after graduation because of low expectations, their expectations soon rise and their satisfaction drops as they are better able to assess their professional activities. Several researchers have found a U-shape relationship between age and job satisfaction [7,34]; older workers are likely to attach less importance to professional ambitions within the context of their lives and acquire a growing awareness of occupational activities from which they derive more satisfaction [35]. Alternative explanations for the greater levels of satisfaction reported by older workers may be that they adapt better than their younger counterparts to the mechanisms and working conditions of their employers [36] or that their seniority entitles them to privileges, such as autonomy and occupational prestige, not commonly experienced by younger workers [37].

The second personal characteristic analyzed was marital status. Decisions to work or not to work, especially for women, and how much or where to work, along with the extent to which work is a source of satisfaction, are intertwined with decisions to remain single, get married, or dissolve a marriage. Family and career choices are associated with each other [38]. Marriage may alter wages and hours worked via its effect on pharmacists’ time horizon, investment in human capital, motivation, or by creating a compensating differential between pleasant jobs that yield satisfaction and unpleasant but better paying jobs. In Finland, Napari [39] finds that women suffer stronger wage losses compared to men, and Edinaldo and Elmslie [40] report that married women participate less in the workforce and supply considerably less labor. In this paper, being married was measured with a dichotomous variable.

Multiple studies show that pharmacists’ professional satisfaction varies positively with earnings [16,17,18,37,41,42,43]. Of all the rewards offered by employers in return for their employees’ labor, pay is the critical variable used as an incentive for worker performance and retention [44,45], so it is not surprising that this is one of the most, if not the most, important determinant of professional satisfaction. Wage-and-salary earnings were identified here with linear and quadratic components; the linear component showed the direction of the effect of earnings on career satisfaction, while the quadratic component showed the rate of change of such effect.

An excessive workload is one of the main sources of pharmacists’ dissatisfaction with their profession [17,46,47]. It generates stress, and eventually burnout, among practitioners and is associated with medication dispensing errors [48,49,50]. In this study, the average number of hours worked per week was used as a proxy for workload. This variable was conceptualized with a linear component measuring the direction of the effect of work hours on career satisfaction and a quadratic component measuring the rate of change.

One of the job-related preferences covariates was the main type of professional activity performed by the pharmacist. Despite the role diversification experienced in recent years, most pharmacists complain that the bulk of their time is spent dispensing medications instead of counseling and clinical activities [51,52]. There is evidence that dispensing is less palatable compared to other roles [17,53]. The main role of the pharmacist was measured here with a dichotomous variable indicating dispensing medications.

The other job-related preferences covariate was commuting time. Traveling to and from work involves real expenses, including an opportunity cost, which condition place of employment, earnings, and other factors related to professional satisfaction [54,55]. This variable is particularly important for women, who traditionally devote more time than men to household work and childrearing activities. When searching for jobs, women are more likely to trade proximity to home or affordable childcare centers for wages, promotions, and/or professional fulfilment [56,57]. Proximity to work was measured in minutes spent on a one-way commute.

The means of male and female pharmacists’ reported levels of career satisfaction and the variables hypothesized to influence these levels were compared using the *t* statistic with an uneven number of observations. Then, ordinary least-squares coefficients were estimated. The least-squares coefficients were designed to show the change in career-satisfaction levels for each one-unit change in the independent variables. After practitioners’ career-satisfaction levels were determined separately for men and women as functions of two personal characteristics (age and marital status), earnings and hours of work (each with linear and quadratic components), and two other job-related variables (main role as a pharmacist and commute time), elasticities were estimated to assess the responsiveness of career-satisfaction levels to the covariates identified as continuous variables. These elasticities, calculated at the means of the continuous variables, measured the ratio of the percentage change in career satisfaction levels to an infinitesimal percentage change in the covariate. Elasticity scores with an absolute value below one were considered to be moderately responsive.

Finally, nine current job-satisfaction indices were added (for men and women separately) to the covariates identified in the initial equations to assess the effect of each of these short-term perceptions on career satisfaction, which is a lifetime construct. The job-satisfaction indices were added one at a time, and independently of one another, to avoid multicollinearity, for a total of ten equations for each gender. The job-satisfaction covariates were stress, job security, availability of advancement opportunities, scheduling flexibility, autonomy, perception of fairness in the workplace, job atmosphere, perceived supervisor’s support, and rating of coworkers. These were assessed by respondents along the same 0–10 intensity scale used to assess career satisfaction. This procedure posed the advantage of response homogeneity, as practitioners expressed their perceptions regarding various facets of their current job following a common measurement standard.

### 2.3. Model

Within each gender, pharmacists’ self-appraised career satisfaction levels were estimated using ordinary least squares as follows:S*_ij(k)_* = α*_ki_* + W*_ijl_*β*_lki_* + X*_ijl_*φ*_lki_* + Y*_ijl_*θ*_lki_* + Z*_ijk_*λ*_ki_* + u*_ijk_*(1)
where 

S*_ij(k)_* was a vector of career-satisfaction levels reported by the *j*th respondent of the *i*th gender. The estimated value changed with the *k*th equation, which is why the *k* subscript appears in parentheses;

W*_ijl_* was a matrix of values of the two personal characteristics (age, in years, and marital status, measured as a dummy variable for being married) reported by the *j*th respondent of the *i*th gender;

X*_ijl_* was a matrix of the annual wage-and-salary earnings (in USD 100,000s) and average workweek (in hours), each with linear and quadratic components, reported by the *j*th respondent of the *i*th gender;

Y*_ijl_* was a matrix of values of the two job-related preferences (dispensing medications as main role, measured as a dummy variable, and one-way commute time, in minutes) reported by the *j*th respondent of the *i*th gender;

Z*_ijk_* was a diagonal matrix of values of the nine job-satisfaction covariates (each measured along a 0–10 scale) reported in the *k*th equation by the *j*th respondent of the *i*th gender;

α*_ki_* was the least-squares constant term estimated in the *k*th equation for the *i*th gender;

β*_lki_* was a vector of the two least-squares personal-characteristics coefficients (one coefficient per covariate) estimated in the *k*th equation for the *i*th gender;

φ*_lki_* was a vector of the four least-squares coefficients for earnings and workweek (each with linear and quadratic components) estimated in the *k*th equation for the *i*th gender;

θ*_lki_* was a vector of the two least-squares job-related preferences coefficients (one coefficient per covariate) estimated in the *k*th equation for the *i*th gender;

λ*_ki_* was a vector of the nine least-squares job-satisfaction coefficients (one coefficient per covariate appearing in a separate equation) estimated in the *k*th equation for the *i*th gender;

u*_ijk_* was a vector of normally and independently distributed stochastic disturbance (random error) terms, with mean zero and variance σ*_ijk_*^2^, pertaining to the *j*th respondent of the *i*th gender in the *k*th equation;

and where

*i* = 1 for men and *i* = 2 for women;

*j* = 1, …, n*_ij_*;

*k* = 0, …, 9 for the equations estimated for each gender;

*l* = 1, 2 for personal characteristics, *l* = 1, …, 4 for earnings and workweek (each with linear and quadratic components), and *l* = 1, 2 for job-related preferences; and

n*_ij_* was the number of respondents in the *i*th gender.

## 3. Results

Of the 2400 questionnaires mailed to potential participants, 139 packets were returned undelivered for various reasons. A total of 737 pharmacists participated in the study by providing answers to all relevant questions for a response rate of 32.6%. The number of observations compared favorably with those reported by similar undertakings in different parts of the world [15,18,23,24,30,31,43,47,58,59]. Of the 737 participants, 57.3% were men and 42.7% were women.

### 3.1. Gender Comparisons of Means

The means and standard deviations of the levels of career satisfaction reported by male and female pharmacists, and the variables hypothesized to influence them, are presented in Table 1. On average, women exhibited 16.0% higher levels of satisfaction with their professional lives than men, and the difference was statistically significant. They also were younger and reported a lower percentage of being married. Compared to women, men earned 11.0% higher levels of annual wages and salaries, and worked, on average, more hours per week. Neither job-related variable showed significantly different means for men and women.

The values of the means and standard deviations of the nine current-job satisfaction indices in the model are presented in Table 2. With the exception of autonomy and fairness in the workplace, all reported indices were statistically different for male and female practitioners, with female pharmacists reporting in all indices higher levels of satisfaction than their male counterparts. Both genders coincided in the ordering of the three highest scores (rating of coworkers, stress, and job atmosphere) as well as the three lowest scores (availability of advancement opportunities, supervisor’s support, and scheduling flexibility).

### 3.2. Estimated Equations

The estimated least-squares coefficients, standard errors, and statistical significance of the 20 equations (ten equations for each gender) are presented in Table 3 for male pharmacists and Table 4 for female pharmacists. In the initial two equations (i.e., without any job-satisfaction index as a covariate), the coefficients for age, marital status, annual earnings, and average workweek were statistically significant for men but not for women; conversely, the coefficients for the job-related preferences covariates were significant for women but not for men. The *F* ratios were significant in both equations and the adjusted R^2^ values were similar for both genders.

The results suggested that age was associated with higher levels of career satisfaction for male pharmacists. Specifically, an additional year of age was associated with an average of 0.033 more points in the career-satisfaction scale. The estimated age elasticity of career satisfaction was 0.309 (see Table 5), meaning that, for example, a 10% increase in age brought about a 3.09% increase in the career-satisfaction scale. The results also suggested that married male pharmacists were significantly more satisfied with their professional activities than their non-married counterparts; they scored, on average, 1.11 points higher in the career-satisfaction scale compared to male pharmacists who were not married.

Continuing with the initial equation for male pharmacists, the estimated annual earnings linear coefficient was positive while the quadratic coefficient was negative, indicating that career-satisfaction levels rose with earnings at a decreasing rate. The estimated annual earnings of career-satisfaction elasticity value (at the means of the variables) was 0.190, which was congruent with this statement. The estimated average workweek linear coefficient for men in the initial equation was negative while the quadratic coefficient was positive, indicating that with additional hours of work, career-satisfaction levels first dropped and eventually reached bottom, increasing afterwards. At its mean, the average workweek elasticity of career satisfaction was estimated to be 0.079.

The only coefficients that were significant in the initial equation for female practitioners were those of the job-related preferences covariates, and both were negative. Female pharmacists whose main role was dispensing medications scored, on average, 1.16 fewer points in the career-satisfaction scale than their counterparts whose main role was other than dispensing. Commuting also affected women’s career satisfaction adversely; on average, each additional minute of commute time reduced career-satisfaction levels by 0.024 points. The commute-time elasticity of career satisfaction was estimated to be −0.082.

The next step in the methodology was to add the nine current-job satisfaction indices, one at a time, to the covariates identified in the initial equation. The estimated least-squares coefficients, standard errors, and statistical significance also are shown in Table 3 for male pharmacists and Table 4 for female pharmacists. The coefficients for both genders of all nine variables were positive and statistically significant, and in every equation the adjusted R^2^ value increased notably with the addition of the job-satisfaction indices. All but one of the job-satisfaction elasticity values (see Table 6), computed at the means of each index, were higher for male than female practitioners; the only exception was availability of advancement opportunities.

## 4. Discussion

Although pharmacists’ career satisfaction is not a novel concept, virtually no empirical work has been conducted in the last ten years on the long-term manifestations of professional contentment. Job satisfaction, its short-term counterpart, has been researched thoroughly, especially in the United States, Canada, England, and the rest of Europe, where commonalities as well as differences in the practice of the profession abound. More recently, however, the exploration has been extended worldwide [59]. This study was the first attempt to link both the long-term and short-term dimensions.

Three major generalizations may be formulated from the probe into pharmacists’ career satisfaction conducted here. First, female practitioners exhibited significantly higher levels of contentment with their careers than their male counterparts. Moreover, women scored higher than men in every index of current-job satisfaction, and all but two of these differences were statistically significant. Thus, compared to their male peers, female pharmacists seemed to derive greater contentment from their professional activities both in the long run (i.e., career satisfaction) and the short run (i.e., satisfaction with facets of current job). This generalization is consistent with other findings in the field of pharmacy [22,23,24] as well as other fields [60,61,62]. It also is consistent with the so-called paradox of the contented female worker, namely, that women’s reported career- and job-satisfaction levels are uniformly greater than the levels reported by men, despite women earning lower levels of income for comparable work, a situation that would be expected to limit satisfaction with one’s professional activities in both the short run and the long run [63,64,65].

The second major generalization of this paper is that the effect of the explanatory variables in the model was altogether different for men and women. The career-satisfaction levels of female pharmacists were not affected significantly by age, marital status, annual earnings, or average workweek, although these covariates systematically influenced male pharmacists’ satisfaction with their careers. The absence of statistical significance of the female (but not male) pharmacists’ coefficients for the earnings and workweek covariates lends credence to the contention that women normally assign less importance to the centrality of market work in their lives, and end up working in the labor market fewer hours and earning lower income levels than men because they usually work relatively more at home [66,67,68,69]. According to this view, men tend to prefer careers in which monetary factors such as pay and overtime hours are emphasized, regardless of type of work, while women are likely to seek jobs that are closer to home in which they may develop their clinical (i.e., non-dispensing) skills.

The empirical evidence suggests that men enjoy their lifetime professional activities more as they grow older, which is compatible with the findings by Ayele, Hawulte, Feto, Basker and Bacha [17] and Iqbal and Iqbal [31]. The estimated function between age and satisfaction was monotonic; there was no evidence of a U-shape relationship, as a quadratic age term included in an earlier regression equation lacked significance for both men and women. The influence of age on career satisfaction was not only statistically significant, but also noteworthy; the age elasticity was moderately inelastic. In addition, married male practitioners seemed to enjoy their professional activities more than their non-married peers.

The empirical evidence further suggests that as annual wage-and-salary earnings rose, the levels of satisfaction reported by male pharmacists increased at a decreasing rate, which conformed to expectations [18,37,43]. According to the estimated coefficients, career-satisfaction levels increased with earnings until reaching a maximum at USD 517,350, which was outside the range of earnings consistent with the data set; for all practical purposes, then, the levels of career satisfaction would never go down as the level of income increased.

The estimated coefficients indicated that as average workweek, a proxy for workload, increased, male pharmacists’ career-satisfaction levels dropped, reached a minimum at approximately 36.1 h, and then increased at an increasing rate. There results conformed to expectations only partially, as they showed a negative association between the number of hours worked and career satisfaction for male pharmacists who worked part time, usually defined up to 36 h per week [70], but not for those who worked full time. A plausible explanation might be that full-time, male pharmacists welcomed the opportunity to work more hours as a means toward higher wages, which might have been a more powerful source of satisfaction than leisure. The fact that earnings were more career-satisfaction elastic than average workweek is compatible with this explanation.

The coefficients for both job-related preferences covariates were significant for female but not male pharmacists. Female practitioners whose main role was dispensing medications exhibited substantially lower levels of career satisfaction than their counterparts whose main role was not dispensing, which was consistent with pharmacists’ wishes to shift from primarily drug distribution activities to more clinical roles, reported by Lau et al. [53] in Hong Kong and Ayele et al. [17] in Ethiopia. Also substantial was the negative influence of commuting time on career satisfaction. Although the elasticity value of this covariate was rather low, the empirical evidence reflected the additional burden of commuting time on women who, in addition to work, are traditionally responsible in the eyes of society for household work and child-rearing activities. This finding is congruent with those by Herbst and Barnow [71], who reported that women’s labor supply was responsive to the geographical distribution of childcare facilities, and Black [66], who concluded that women with young children were sensitive to longer commutes.

The third major generalization is that pharmacists’ satisfaction with different facets of their current job, a short-term perception, influenced greatly their long-term career satisfaction. Not only were all coefficients statistically significant for male and female practitioners, but the elasticity values reflected responsiveness of career satisfaction to the job-satisfaction indices. This responsiveness was generally greater for male than for female pharmacists. The greatest job-satisfaction elasticity values for men were rating of coworkers, fairness in the workplace, and job atmosphere; curiously, two of these indices, rating of coworkers and job atmosphere, were reported by female respondents in the top three sources of job satisfaction (see Table 2). The coefficients of multiple determination, which measured the percentage variation in career satisfaction levels explained by the equation covariates, increased considerably with the inclusion of each current-job satisfaction index.

The coefficients of all job-satisfaction covariates conformed to expectations except the stress index for both men and women, which portrayed a positive influence on career satisfaction and contravened earlier findings [17,72,73,74,75]. Perhaps the positive coefficients suggested that while within a specific work setting greater stress might reduce contentment, pharmacists generally accepted that more challenging and satisfying jobs are accompanied by more stress [22]. As Gaither et al. [76] have pointed out, the practice of pharmacy can be simultaneously satisfying and highly stressful.

The empirical evidence showed a strong link between the job-satisfaction variables, which provided short-term perspectives of current employment, and the long-term assessment of professional contentment. Since both types of indicators were measured along the same 0-10 scale, the connection was easy to elucidate. For example, other things equal, an additional point in the job-security satisfaction scale resulted in a 0.38 point increase in the career-satisfaction scale of male pharmacists and a 0.32 point increase in the career-satisfaction scale of female pharmacists. Similarly, an additional point of satisfaction with availability of advancement opportunities in one’s place of employment brought about, on average, 0.21 more points for male pharmacists and 0.32 more points for female pharmacists in their career-satisfaction scale. In virtually all cases, the size of the men’s job-satisfaction index coefficients exceeded the size of the women’s coefficients.

Beyond the covariates reported above, the influence on pharmacists’ career satisfaction of other covariates, not shown, was tested. These covariates measured, as indicators of personal characteristics, ethnic identification, number of children, type of academic degree, whether or not respondent completed a residency or fellowship, whether or not respondent held a specialty board certification, and whether or not (if married) the spouse worked. In addition, type of primary practice site (retail, hospital, other), location of primary practice site (large city, small city or town, rural), experience as a registered pharmacist, longevity in current job, and whether or not respondent served primarily a specific patient population were tested as indicators of job-related preferences. None of these covariates yielded a statistically significant coefficient for either gender. Their contribution to explaining variation in career satisfaction levels was so feeble that the other least-squares coefficients appearing in the reported equations did not change appreciably when they were deleted. Thus, they were excluded.

## 5. Limitations

In interpreting the results of this paper, one must take into account several limitations inherent to the study. First, self-reported data were used, which by their very nature were subject to validity and reliability concerns even though the questionnaire was tested prior to being mailed to participants. Career satisfaction is subjective, and practitioners’ feelings regarding satisfaction or its determinants may change frequently. The study used a cross-sectional survey, which was inadequate to measure the variation of such perceptions over time, especially when the composition and functions of the pharmacist workforce are evolving rapidly. The role of the pharmacist currently embraces collaborative team-based care, medication therapy management and reconciliation, preventive care services, and patient education and behavioral counseling, all of which are likely to affect perceptions of career satisfaction. Furthermore, the proliferation of pharmacy schools and graduates leading to increasing competition for a limited (or even shrinking) number of jobs, as well as the growing shift toward mailed-order distribution of prescription drugs, also may be relevant factors. Future research ought to include longitudinal data in an attempt to fully comprehend career satisfaction over the life cycle.

Since responses remained anonymous, there was no way to corroborate that the questionnaires were received and filled by the intended respondents, which was a crucial validity issue. Also pertaining to validity was the fact that the accuracy of neither wage-and-salary earnings nor number of hours worked per week was checked with employers. In addition, potential measurement biases in the model variables need be recognized. Kankaanranta and Rissanen [77] and Polgreen*,* et al. [78] have warned about a sample selection bias that occurs when the data include only practitioners who have made the decision to work; if a substantial number of pharmacists chose not to respond to the survey because they were not working, either voluntarily (e.g., retired, raising children, etc.) or not (e.g., unemployed), the indicators reported here might be biased, although given the sample size, they probably were consistent. In addition, Evers*,* et al. [79] have pointed out that the ordinary least squares estimation technique might suffer from serious shortcomings, while the presence of multicollinearity generated by the relationship between annual earnings and the number of hours worked might have inflated the standard errors of the female pharmacists’ coefficients, thus explaining partially the lack of statistical significance of these two variables.

No incentives such as monetary compensation or raffle prizes were used to motivate pharmacists to participate in the survey; these incentives might have altered the number of respondents as well as the nature of the responses. Potential biases from the omission of covariates also need be recognized. While several covariates were tried and discarded in preliminary equations, the effect of others was never explored. For example, wage-and-salary earnings might have portrayed an incomplete view of the influence of income on professional satisfaction, since they did not include other sources of professional income or total household income; other variables not considered were variations in tax rates, differences in price levels throughout the country, job-satisfaction indicators pertaining to the meaningfulness of respondents’ jobs, personality traits, and non-career life satisfaction.

## 6. Conclusions

Understanding the presence (or absence), composition, and sources of career satisfaction experienced by pharmacists may help employers configure and/or alter workplace conditions to more adequately meet practitioners’ needs and expectations, increase their productivity, and allocate human resources more efficiently. More satisfied workers generally see their organization in a positive manner and are appreciative toward their employer for providing them with a professional environment that makes them happy. They tend to invest in firm-specific human capital, are less likely to leave their jobs voluntarily, and are less prone to abandon their profession than are less satisfied workers. In this study, men and women responded differently to stimuli, so a uniform set of incentives and disincentives may not be equally effective for both genders. Initiatives perceived by male pharmacists as increasing satisfaction may be adversely perceived by female pharmacists, and vice versa.

Despite its limitations, this study has been successful in conceptualizing and measuring pharmacists’ career satisfaction, identifying important determinants, and comparing its prevalence between male and female practitioners. The study also linked short-term opinions on various facets of pharmacists’ current jobs to a long-term assessment of contentment with their professional activities. The approach was novel, so findings should be regarded as preliminary in nature. Further research is needed to devise new variables and functional relationships that shed additional light into the nature and mechanisms of this important component of the pharmacist workforce, not only in the United States but also in other parts of the world

## Figures and Tables

**Table 1 pharmacy-09-00173-t001:** Mean and standard deviation (in parentheses) values of the pharmacists’ career-satisfaction index and related variables in the initial equation, by gender.

Variables	Mean and Standard Deviation Values	
	Men	Women
Number of observations	422	315
Career satisfaction index (0–10 scale) [C]	6.024 * (3.028)	6.990 * (2.421)
PERSONAL CHARACTERISTICS [W]		
Age (years)	56.0 * (11.5)	47.6 * (9.8)
Marital status: Married (%)	80.9 ^§^ (15.5)	72.9 ^§^ (19.8)
EARNINGS AND HOURS [X]		
Annual wage-and-salary earnings (USD)	116,807.8 ^§^ (83,575.6)	105,246.6 ^§^ (31,361.4)
Average workweek (hours)	40.0 * (10.9)	37.6 * (10.5)
JOB-RELATED PREFERENCES [Y]		
Main role: Dispensing medications (%)	73.3 (19.6)	71.3 (20.5)
Commute time (minutes)	22.6 (20.5)	23.8 (18.8)

* Significantly different from each other (*p* ≤ 0.01). ^§^ Significantly different from each other (*p* ≤ 0.05).

**Table 2 pharmacy-09-00173-t002:** Mean and standard deviation (in parentheses) values of job-satisfaction indices related to pharmacists’ career satisfaction, by gender.

Variables	Mean and Standard Deviation Values	
JOB-SATISFACTION INDICES [Z]	Men	Women
Stress (0–10 scale)	6.126 * (3.003)	6.953 * (2.520)
Job security (0–10 scale)	5.861 ^§^ (3.173)	6.361 ^§^ (2.708)
Advancement opportunities (0–10 scale)	3.246 ^†^ (2.961)	3.689 ^†^ (3.069)
Scheduling flexibility (0–10 scale)	5.472 * (3.203)	6.140 * (3.092)
Autonomy (0–10 scale)	5.809 (2.832)	6.108 (2.603)
Fairness in the workplace (0–10 scale)	5.995 (3.063)	6.171 (2.721)
Job atmosphere (0–10 scale)	6.044 ^†^ (2.918)	6.403 ^†^ (2.638)
Supervisor’s support (0–10 scale)	5.213 * (3.333)	5.969 * (3.128)
Rating of coworkers (0–10 scale)	6.372 * (2.939)	6.990 * (2.447)

* Significantly different from each other (*p* ≤ 0.01). ^§^ Significantly different from each other (*p* ≤ 0.05). ^†^ Significantly different from each other (*p* ≤ 0.10).

**Table 3 pharmacy-09-00173-t003:** Estimated least-squares coefficients, their standard errors (in parentheses), and (two-tail) levels of significance of covariates in the model for male pharmacists (*i* = 1).

Equat. #	Least-Squares Coefficients and Their Standard Errors																		*F* Ratio	Adjust. R^2^
	Const. Term (α_11_)	Pers. Characteristics		Annual Earnings		Average Workweek		Job-Related Prefer.		Job-Satisfaction Indices										
		Age (β_11_)	Married (β_12_)	Linear (φ_11_)	Squared (φ_12_)	Linear (φ_13_)	Squared (φ_14_)	Disp. (θ_11_)	Comm. (θ_12_)	Stress (λ_11_)	Security (λ_12_)	Opport. (λ_13_)	Flexib. (λ_14_)	Auton. (λ_15_)	Fairness (λ_16_)	Atmos. (λ_17_)	Support (λ_18_)	Cowrk. (λ_19_)		
*k* = 0	3.3644	0.0332 ^§^ (0.0145)	1.1105 * (0.4028)	1.2689 ^§^ (0.5060)	−0.1226 * (0.0467)	−0.1106 ^†^ (0.0663)	0.0015 ^†^ (0.0008)	0.4437 (0.3810)	−0.0031 (0.0077)										3.48 *	0.050
*k* = 1	3.2032	0.0301 ^§^ (0.0144)	1.0538 * (0.3978)	1.2164 ^§^ (0.4995)	−0.1222 * (0.0461)	−0.1253 ^§^ (0.0656)	0.0016 ^§^ (0.0008)	0.3682 (0.3765)	−0.0022 (0.0076)	0.1723 * (0.0520)									4.39 *	0.076
*k* = 2	2.0666	0.0206 (0.0135)	0.8261 ^§^ (0.3742)	0.8565 ^†^ (0.4707)	−0.0836 ^†^ (0.0434)	−0.0899 (0.0614)	0.0012 ^†^ (0.0008)	0.5343 (0.3524)	0.0030 (0.0072)		0.3766 * (0.0475)								10.61 *	0.188
*k* = 3	2.5544	0.0367 * (0.0143)	0.9848 ^§^ (0.3966)	1.1234 ^§^ (0.4986)	−0.1078 ^§^ (0.0460)	−0.1140 ^†^ (0.0651)	0.0017 ^§^ (0.0008)	0.5455 (0.3748)	−0.0025 (0.0076)			0.2087 * (0.0542)							4.84 *	0.085
*k* = 4	2.9927	0.0144 (0.0134)	0.8453 ^§^ (0.3675)	0.9175 ^§^ (0.4619)	−0.0877 ^§^ (0.0426)	−0.1018 ^†^ (0.0603)	0.0013 ^†^ (0.0007)	0.3123 (0.3467)	−0.0023 (0.0070)				0.3993 * (0.0454)						12.34 *	0.215
*k* = 5	2.1590	0.0280 ^§^ (0.0137)	0.8191 ^§^ (0.3797)	1.1204 ^§^ (0.4748)	−0.1017 ^§^ (0.0439)	−0.1278 ^§^ (0.0622)	0.0017 ^§^ (0.0008)	0.4844 (0.3572)	0.0006 (0.0073)					0.3834 * (0.0532)					9.34 *	0.168
*k* = 6	1.7404	0.0228 ^†^ (0.0128)	0.9685 * (0.3528)	0.4872 (0.4488)	−0.0521 (0.0414)	−0.1058 ^†^ (0.0581)	0.0016 ^§^ (0.0007)	0.2926 (0.3344)	0.0029 (0.0068)						0.4795 * (0.0453)				16.56 *	0.274
*k* = 7	1.8799	0.0190 (0.0131)	0.9991 (0.3620)	0.8696 ^†^ (0.4565)	−0.0810 ^†^ (0.0422)	−0.1100 ^†^ (0.0596)	0.0015 ^§^ (0.0007)	0.3966 (0.3422)	0.0004 (0.0070)							0.4555 * (0.0484)			13.65 *	0.234
*k* = 8	2.4867	0.0245 ^†^ (0.0141)	0.8511 ^§^ (0.3916)	1.0277 ^§^ (0.4957)	−0.0986 ^§^ (0.0454)	−0.0913 (0.0639)	0.0013 ^†^ (0.0008)	0.4019 (0.3710)	−0.0059 (0.0075)								0.2918 * (0.0461)		7.99 *	0.147
*k* = 9	2.5450	0.0153 (0.0131)	0.9842 * (0.3631)	0.4317 (0.4621)	−0.0422 (0.0427)	−0.1059 ^†^ (0.0593)	0.0014 ^§^ (0.0007)	0.2015 (0.3427)	−0.0021 (0.0069)									0.4593 * (0.0484)	13.97 *	0.239

* Coefficient statistically significant (*p* ≤ 0.01). ^§^ Coefficient statistically significant (*p* ≤ 0.05). ^†^ Coefficient statistically significant (*p* ≤ 0.10).

**Table 4 pharmacy-09-00173-t004:** Estimated least-squares coefficients, their standard errors (in parentheses), and (two-tail) levels of significance of covariates in the model for female pharmacists (*i* = 2).

Equat. #	Least-Squares Coefficients and Their Standard Errors																		*F* Ratio	Adjust. R^2^
	Const. Term (α_21_)	Pers. Characteristics		Annual Earnings		Average Workweek		Job-Related Prefer.		Job-Satisfaction Indices										
		Age (β_21_)	Married (β_22_)	Linear (φ_21_)	Squared (φ_22_)	Linear (φ_23_)	Squared (φ_24_)	Disp. (θ_21_)	Comm. (θ_22_)	Stress (λ_21_)	Security (λ_22_)	Opport. (λ_23_)	Flexib. (λ_24_)	Auton. (λ_25_)	Fairness (λ_26_)	Atmos. (λ_27_)	Support (λ_28_)	Cowrk. (λ_29_)		
*k* = 0	9.2317	−0.0182 (0.0157)	0.0380 (0.3432)	0.2064 (2.0832)	0.3131 (0.9486)	0.0071 (0.0693)	−0.0006 (0.0008)	−1.1576 * (0.3498)	−0.0241 * (0.0080)										3.41 *	0.071
*k* = 1	8.4794	−0.0212 (0.0157)	0.1167 (0.3452)	0.7835 (2.0884)	0.0529 (0.9513)	−0.0112 (0.0694)	−0.0004 (0.0008)	−1.1974 * (0.3513)	−0.0240 * (0.0080)	0.1439 ^§^ (0.0602)									3.60 *	0.086
*k* = 2	6.0546	−0.0070 (0.0148)	0.2575 (0.3208)	0.8853 (1.9411)	−0.0835 (0.8846)	0.0033 (0.0644)	−0.0005 (0.0007)	−0.9234 * (0.3318)	−0.0169 ^§^ (0.0075)		0.3170 * (0.0546)								7.37 *	0.186
*k* = 3	6.5741	0.0003 (0.0143)	0.1110 (0.3211)	−1.8187 (1.9610)	0.9231 (0.8864)	0.0766 (0.0653)	−0.0012 (0.0007)	−0.8910 * (0.3283)	−0.0186 ^§^ (0.0075)			0.3198 * (0.0495)							8.14 *	0.204
*k* = 4	6.4786	−0.0212 (0.0144)	0.1274 (0.3138)	0.6155 (1.9030)	0.0663 (0.8669)	0.0318 (0.0634)	−0.0007 (0.0007)	−0.9202 * (0.3251)	−0.0212 * (0.0073)				0.2972 * (0.0457)						8.47 *	0.211
*k* = 5	6.1611	−0.0201 (0.0142)	0.1150 (0.3091)	−0.0596 (1.8658)	0.3976 (0.8486)	0.0133 (0.0621)	−0.0006 (0.0007)	−0.7276 ^§^ (0.3233)	−0.0088 (0.0074)					0.4202 * (0.0548)					10.34 *	0.253
*k* = 6	5.1826	−0.0167 (0.0140)	0.0232 (0.3056)	1.5833 (1.8542)	−0.2314 (0.8431)	0.0291 (0.0615)	−0.0009 (0.0007)	−0.7808 ^§^ (0.3183)	−0.0116 (0.0072)						0.3922 * (0.0512)				10.60 *	0.256
*k* = 7	6.1814	−0.0244 ^†^ (0.0141)	0.1793 (0.3088)	0.7687 (1.8717)	0.0094 (0.8525)	0.0237 (0.0623)	−0.0008 (0.0007)	−1.0263 * (0.3172)	−0.0141 ^†^ (0.0073)							0.3831 * (0.0531)			9.69 *	0.237
*k* = 8	7.3416	−0.0255 ^†^ (0.0145)	0.0672 (0.3188)	−0.3183 (2.0079)	0.6585 (0.9326)	0.0254 (0.0648)	−0.0009 (0.0007)	−0.9338 * (0.3304)	−0.0119 (0.0075)								0.2853 * (0.0468)		8.18 *	0.207
*k* = 9	6.6006	−0.0154 (0.0143)	−0.1369 (0.3139)	−0.9677 (1.9016)	0.6428 (0.8642)	0.0410 (0.0632)	−0.0009 (0.0007)	−1.1786 * (0.3197)	−0.0190 * (0.0073)									0.3760 * (0.0560)	8.82 *	0.218

* Coefficient statistically significant (*p* ≤ 0.01). ^§^ Coefficient statistically significant (*p* ≤ 0.05). ^†^ Coefficient statistically significant (*p* ≤ 0.10).

**Table 5 pharmacy-09-00173-t005:** Career-satisfaction elasticity values estimated at the means of the continuous variables in the initial equation, by gender.

Continuous Variables in the Initial Equation	Career-Satisfaction Elasticity Values	
	Men	Women
Age	0.309	a
Annual wage-and-salary earnings	0.190	a
Average workweek	0.079	a
Commute time	a	−0.082

a: Least-squares coefficients not statistically significant (*p* ≤ 0.10).

**Table 6 pharmacy-09-00173-t006:** Career-satisfaction elasticity values of the nine current job-satisfaction indices estimated at the means of the variables, by gender.

Current Job-Satisfaction Indices	Career-Satisfaction Elasticity Values	
	Men	Women
Stress	0.175	0.143
Job security	0.366	0.288
Advancement opportunities	0.112	0.169
Scheduling flexibility	0.363	0.261
Autonomy	0.370	0.367
Fairness in the workplace	0.477	0.346
Job atmosphere	0.457	0.351
Supervisor’s support	0.253	0.244
Rating of coworkers	0.486	0.376

## Data Availability

Unidentified data were gathered by the authors from responses by participants to a questionnaire made available via postal mail and electronic mail.

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
