# Peer review of "Gender and Pharmacists’ Career Satisfaction in the United States"

_pharmacy, 2021, doi:10.3390/pharmacy9040173_

Round 1
Reviewer 1 Report
This article examines career satisfaction among US pharmacists, specifically aiming to identify any differences among the genders. I must commend the authors on a thorough, well designed study to achieve this objective.
It is clear the authors are well-versed and provided substantial literature evidence in both describing the background and describing their selection of variables. In the methods section, the authors referenced the survey used was validated; I would like to see the reference for this validation and provide a copy of the survey as a supplement if possible.
The selection of data variables was well described. I do feel like the reader would need a substantial understanding of predictive modeling, beta coefficients, and their application to understand the results. Given the broad audience of this publication, I suggest describing in text how the variables behaved in each model and removing tables 3 and 4. While this is the crux of the objective, as presented, the results depends criticially on the education of the reader. Either the final model or major differences you would like to highlight can be a table or just describe the relationships in text. I think this would help the reader synthesize the information, rather than provided the detailed data.
Overall, I think the interpretation, discussion and conclusions are fair to the research objective and is a high quality article. Thank you for sharing your work!
Reviewer 2 Report
A well designed study which adress an important issue of quality of life of a health professional, the pharmacists. To my knowledge is not very often studied.
In the discussion it would be interesting to introduce comparisons with Canada and England (the practice is different in south Europe where pharmacy are mainly privatly owned).
It might be interesting to discuss how changes could be implemented according to the findings
Reviewer 3 Report
INTRODUCTION:
The introduction could be strengthened in the following areas:
- What was the rationale for the study? Was there any relevance in the current environment that led to this study? Was there a framework used? How is this different than the Pharmacist Workforce Study? Why were the particular analyses conducted (gender?). the introduction needs to specify the reasoning for choosing these variables.
METHODS:
- The biggest issue I had with this paper was the lack of clarity in the survey questions…I read it couple times and still could not ascertain if it was just a question or had comment section etc.
- Survey: 0-10 intensity scale, 10 is the greatest level of satisfactionàdoesn’t clarify how many questions in the survey. They mentioned 10 questions in job-satisfaction part, but not sure in career satisfaction part.
- “This scale provided more room for discrimination in participants’ response than is normally provided by a Likert scale. The rather narrow range of options ensured the adequacy of the mean as a measure of central tendency, since there was no possibility of outliers. This type of indicator has been applied successfully in previous studies”àcontradiction? They have 10 scales while common Likert scale is less than 10, so they’re saying the larger scale is better than typical Likert scale but narrow range provide better central tendency? (Line 76)
- How did they write the questionnaire, did they use directly from other literature, or did they review from other literatures and edited/modified by themselves?
- “The survey questionnaire, previously validated and exclusively prepared for this and other workforce studies, was mailed by the authors in March 2012” (Line 101)
- No Inclusion/exclusion criteria? How did they select valid respondents from the returned survey? What’s the time frame?
RESULTS:
- Gender comparison of means:
- Table 1:
- Age (year): M is 56, F is 47? And the year is how long they have been practice?
- Also, they mentioned female participants were younger and reported a lower percentage of being marriedà but 19.8% was higher than 15.55%? (Line 239)
- Table 1:
Discussion:
- Comparisons between gender was followed with reasonable explanations with every result
Other suggestions:
- How did they test their questionnaire is valid and reliable?
- Suggest having an appendix to include all of the survey questions
- Subtitle for “career-satisfaction” and “job-satisfaction” in the discussion/result section is more reader friendly and looks more organized
Tables:
Could not stand alone – need clearer footnotes.
